

# Iron overload induced death of osteoblasts *in vitro*: involvement of the mitochondrial apoptotic pathway

Qing Tian[1,*], Shilei Wu[1,*], Zhipeng Dai[2], Jingjing Yang[3], Jin Zheng[4], Qixin Zheng[1] and Yong Liu[1]

[1] Department of Orthopedics, Union Hospital, Tongji Medical College, Huazhong University of Science and Technology, Wuhan, China
[2] Department of Orthopedics, Henan Provincial People's Hospital, Zhengzhou, China
[3] Department of Child Health, Changzhou Maternal and Child Health Care Hospital, Changzhou, China
[4] Department of Neurology, Union Hospital, Tongji Medical College, Huazhong University of Science and Technology, Wuhan, China
[*] These authors contributed equally to this work.

Corresponding authors
Qixin Zheng, zheng-qx@163.com
Yong Liu, 573846007@qq.com

## ABSTRACT

**Background**. Iron overload is recognized as a new pathogenfor osteoporosis. Various studies demonstrated that iron overload could induce apoptosis in osteoblasts and osteoporosis *in vivo*. However, the exact molecular mechanisms involved in the iron overload-mediated induction of apoptosis in osteoblasts has not been explored.

**Purpose**. In this study, we attempted to determine whether the mitochondrial apoptotic pathway is involved in iron-induced osteoblastic cell death and to investigate the beneficial effect of N-acetyl-cysteine (NAC) in iron-induced cytotoxicity.

**Methods**. The MC3T3-E1 osteoblastic cell line was treated with various concentrations of ferric ion in the absence or presence of NAC, and intracellular iron, cell viability, reactive oxygen species, functionand morphology changes of mitochondria and mitochondrial apoptosis related key indicators were detected by commercial kits. In addition, to further explain potential mechanisms underlying iron overload-related osteoporosis, we also assessed cell viability, apoptosis, and osteogenic differentiation potential in bone marrow-derived mesenchymal stemcells(MSCs) by commercial kits.

**Results**. Ferric ion demonstrated concentration-dependent cytotoxic effects on osteoblasts. After incubation with iron, an elevation of intracelluar labile iron levels and a concomitant over-generation of reactive oxygen species (ROS) were detected by flow cytometry in osteoblasts. Nox4 (NADPH oxidase 4), an important ROS producer, was also evaluated by western blot. Apoptosis, which was evaluated by Annexin V/propidium iodide staining, Hoechst 33258 staining, and the activation of caspase-3, was detected after exposure to iron. Iron contributed to the permeabilizatio of mitochondria, leading to the release of cytochrome C (cyto C), which, in turn, induced mitochondrial apoptosis in osteoblasts via activation of Caspase-3, up-regulation of Bax, and down-regulation of Bcl-2. NAC could reverse iron-mediated mitochondrial dysfunction and blocked the apoptotic events through inhibit the generation of ROS. In addition, iron could significantly promote apoptosis and suppress osteogenic differentiation and mineralization in bone marrow-derived MSCs.

**Conclusions**. These findings firstly demonstrate that the mitochondrial apoptotic pathway involved in iron-induced osteoblast apoptosis. NAC could relieved the

oxidative stress and shielded osteoblasts from apoptosis casused by iron-overload. We also reveal that iron overload in bone marrow-derived MSCs results in increased apoptosis and the impairment of osteogenesis and mineralization.

## INTRODUCTION

Iron is an essential element for several cellular and metabolic processes. However, this transition metal also catalyzes the formation of damaging free radicals, leading to the oxidative injury of cellular components. Osteoporosis and fractures occur frequently in patients with disorders associated with iron overload such as thalassemia and hemochromatosis (*Vogiatzi et al., 2006*; *Vogiatzi et al., 2009*; *Kim et al., 2012*; *Wong et al., 2014*). Evidence from numerous studies indicates that iron overload directly exerts detrimental effects on bone metabolism (*Yang et al., 2011*; *Tsay et al., 2010*). Excessive iron accumulation in osteoblast triggers apoptosis which may play essential roles in osteoporosis (*Messer et al., 2009*; *Doyard et al., 2012*). However, the mechanism by which iron induces apoptosis is not fully understood. Therefore, the elucidation of the mechanisms underlying apoptosis and development of therapeutic strategies to block apoptosis in osteoblasts are crucial for treating iron-overload induced osteoporosis.

Apoptosis occurs via two major pathways: the death receptor pathway and mitochondrial pathway. The death receptor pathway is mainly initiated by the ligation of death receptors such as tumor necrosis factor (TNF) and Fas/CD95, in which the recruited caspase 8 acts as a trigger for the activation of caspase 3 and apoptosis (*Fuchs & Steller, 2011*). The mitochondrial pathway, known as another important apoptotic pathway, is activated by various stimuli that induce the dissipation of the mitochondrial membrane and the release of apoptotic factors such as cyto *c* (*Kroemer, Galluzzi & Brenner, 2007*; *Green, Galluzzi & Kroemer, 2014*). After cyto c is released into the cytosol, its initiates the formation of cytochrome c /Apaf-1/ Caspase-9 complex (termed the apoptosome), which causes the activation of Caspase-3, subsequently executing cell apoptosis (*Tait & Green, 2013*). However, the specific apoptotic pathway by which iron induces apoptosis in osteoblasts has not been reported.

Studies have confirmed that redox-active iron in mitochondria is capable of directly catalyzing the formation of deterimental free radicals via Fenton chemistry (*Lill, 2009*; *Dixon & Stockwell, 2014*). Both iron-dependent ROS and high concentrations of labile iron are thought to contribute to mitochondrial ultrastructural damage (*Pietrangelo, 2016*). Therefore, we hypothesize that iron-induced osteoblast apoptosis may be mediated by the mitochondrial apoptotic pathway. Furthermore, we explored the role of the mitochondrial pathway in iron-overload induced osteoblast apoptosis by examining mitochondrial function and apoptosis related key molecules. In additon, in our present study, the protection of NAC on iron overload-induced osteoblasts apoptosis has been investigate.

## MATERIALS & METHODS

### Materials

N-acetyl-cysteine (NAC), Ferric ammonium citrate (FAC), 2′, 7′-dichlorodihy-drofluorescein diacetate (H2DCF-DA), calcein-AM, Hoechst33258, 4′, 6-diamidino-2-phenylindol (DAPI), penicillin, streptomycin, leupeptin, pepstatin A, deaprotinin, phenylmethylsulfonylfluoride (PMSF), and 4-(2-hydroxyethyl)-1-piperazineethane sulfonic acid (HEPES) were purchased from Sigma (St. Louis, MO, USA). Primary antibodies against cytochrome c, bcl-2, bax, cleaved Caspase-3, Glyceraldehyde-3-Phosphate Dehydrogenase (GAPDH) and beta-actin ($\beta$-actin) were purchased from Abcam (Cambridge, UK). AnnexinV–FITC/PI kit was obtained from KeyGen Biotech (Nanjing, China). Fetal bovine serum and alpha-modified Eagle's medium (a-MEM) were purchased from Gibco (Waltham, MA, USA). Cell Mitochondria Isolation Kit, Enhanced Chemiluminescence detection kit, Western and immunoprecipitation (IP) Cell Lysis Kit, Bicinchoninic Acid Protein (BCA) Protein Assay Kit, and 5, 5′, 6, 6′-tetrachloro-1, 1′, 3, 3′-tetraethylbenzimidazolcarbocyanineiodide (JC-1) were purchased from Beyotime (China). Cell Counting Kit-8 (CCK-8) assay kit was purchased from DojinDo (Japan).

### Cell cultures and treatment

MC3T3-E1 osteoblasts (obtained from American Type Culture Collection) were cultured in alpha-modified Eagle's medium supplemented with 10% fetal bovine serum (FBS) (Gibco H; Invitrogen, Grand Island, NY, USA), 50 U/ml penicillin, and 50 mg/ml streptomycin. The medium was changed thrice one week. Cells were cultured to 80–90% confluence, harvested, and seeded at $1 \times 10^4$ cell/cm$^2$ in 96- and 6-well plates.

FAC, which functions as an iron donor, was used to simulate iron overload conditions *in vitro* (*Doyard et al., 2012*; *Zarjou et al., 2010*). We incubated MC3T3-E1 cells with various concentrations of FAC: 25, 50, 100, and 200 μM. Control groups were treated with PBS. After exposure to FAC in the absence or presence of NAC (1 mM), all samples were collected subsequent analyses by flow cytometry, Western Blot, confocal microscopy, and fluorescence microscopy.

Primary bone marrow-derived MSCs were isolated from Sprague-Dawley rats (100–120 g, obtained from the Animal Center of Tong Ji Medical College, Huazhong University of Science and Technology) as previously described (*Meng et al., 2016*). The isolated cells were cultured in 55-m$^2$ dishes in Dulbecco's modified Eagle's medium (HyClone, Logan, UT, USA) with 10% (v/v) fetal bovine serum (FBS) and 100 μg/mL streptomycin and penicillin (Beyotime Institute of Biotechnology, Jiangsu, China) at 37 °C, 5% CO$_2$ atmosphere. The growth medium was changed every 2 days. Primary bone marrow-derived MSCs were grown to confluence and used from passages 3 to 6 throughout the following experiments.

### Measurement of cell viability

The cell counting kit-8 was applied to determine viability of osteoblastic cells and bone marrow-derived MSCs as described previously (*Ding et al., 2012*; *Cai et al., 2015*). After exposure to FAC (25–200 μM) for 24 and 120 h, the mixture solution containing medium (90 μl) and CCK-8 reactant (10 μl) was added to each well of a 96-well plate. Then, the

sample was incubated at 37 °C for 2 h in the dark. Finally, the absorbance at 450 nm was analyzed in a spectrophotometer (Thermo, Waltam, MA, USA).

## Assay of the intracellular labile iron level by flow cytometry and fluorescence microscopy

The labile iron pool (LIP), which refers to the level of intracellular redox-active and chelatable iron, has been implicated in cellular damage by catalyzing excess-production of deterimental free radical. The intracellular LIP was measured by flow cytometry following calcein staining. Briefly, after treatment with FAC (0–200 $\mu$M) for 120 h, the osteoblasts were collected and resuspended in PBS, and subsequently treated with 0.25 $\mu$M calcein-AM in dark at 37 °C for 30 min (*Tenopoulou et al., 2007*; *Glickstein et al., 2005*; *Kaur et al., 2009*). Next, osteoblasts were rinsed twice with serum-free a-MEM, gently resuspended in the medium, and immediately analyzed by flow cytometry using CellQuest analysis software (BD Biosciences, USA). Meanwhile, in order to evaluate the change in intracellular labile iron levels *in situ*, all samples were additionally monitored under a fluorescence microscope.

## Evaluation of intracellular reactive oxygen species

Levels of ROS in osteoblasts were determined with H2DCF-DA, a fluorescent dye, which could be rapidly oxidized into the highly fluorescent compound DCF in the presence of ROS (*Ma et al., 2013*). Following treatment with FAC, the osteoblasts were collected and washed with PBS, subsequently resuspended in serum-free media, and finally treated with 20 $\mu$M H2DCF-DA in dark at 37 °C for 20 min (*Ding et al., 2012*). After incubation, MC3T3-E1 cells were rinsed with serum-free a-MEM thrice, and subsequently the mean fluorescence intensity (MFI) was evaluated with a FACSCalibur flow cytometer (BD, Franklin Lakes, NJ, USA).

## Evaluation of apoptosis by Annexin V-FITC/PI staining

After treatment as described above, osteoblasts and bone marrow-derived MSCs from each sample were stained using the Annexin V–FITC/PI kit, as described previously (*Ding et al., 2012*; *Cai et al., 2015*). Then, flow cytometry was accomplished by flow cytometric analysis (BD, USA). The sample was additionally visualized under a confocal microscope (OLYMPUS FV1000, Japan). The Annexin V+/PI- osteoblasts were considered early apoptotic cells (*Henry, Hollville & Martin, 2013*). The Annexin V+/PI+ osteoblasts were considered late apoptotic cells (*Henry, Hollville & Martin, 2013*).

## Evaluation of apoptosis-related morphologic changes in osteoblasts

To verify apoptosis, Ding (*Ding et al., 2012*; *Cai et al., 2015*) outlined a procedure for ascertaining apoptosis, through the application of Hoechst 33258 dye at 0.5 $\mu$g/mL concentrate, to clarify the fractured and compressed apoptotic nuclei. Following the application of the dye, the nuclei were stained for one hour at 37 °C with all light sources omitted. The nuclei were then given two washes with PBS, so as to remove the Hoechst 33258. Utilising a fluorescence microscope manufactured by Carl Zeiss, the morphology of osteoblast apoptosis was pictured with UV excitation at 350nm, then analysed.

## Measurement of mitochondrial membrane potential (MMP)

The collapsed MMP in osteoblasts was detected by flow cytometry following JC-1 dye staining (Beyotime-biotechnology, China). Briefly, after treatment as described above, the harvested MC3T3-E1 cells were resuspended in a staining solution, which were prepared by admixing serum-free a-MEM (500 μL) and JC-1 staining fluid (500 μL). Subsequently, osteoblasts were incubated with the staining solution for 20 min at 37 °C in the dark. After incubation, osteoblasts were washed thrice with the JC-1 staining buffer (Beyotime Institute of Biotechnology, Jiangsu, China), and resuspended in 500 μL of cell culture medium prior to analysis by flow cytometry. Finally, the ratio of red fluorescence intensity to green fluorescence intensity was calculated, and the results were used to evaluate the change in MMP for each sample (*Ding et al., 2012*).

In order to evaluate the change in MMP *in situ*, osteoblasts were loaded with JC-1 dye after treatment with FAC, as described above (*Ding et al., 2012*). Aggregated JC-1 (red fluorescence) and monomeric JC-1 (green fluorescence) levels were observed using laser scanning confocal microscopy (Olympus FV1000; Olympus, Tokyo, Japan).

## Western blot analysis

After treatment with FAC, as described above, osteoblasts were collected and homogenized in lysis buffer. The details of the Western blotting procedures have been described previously (*Ding et al., 2012*). The monoclonal antibodies used were as follows: anti-cleaved Caspase-3 (1: 2,000), anti-Bax (1 : 1,000), anti-Bcl-2 (1 : 1,000), and anti-Cytochrome c (1 : 1,000). GAPDH and beta-actin were utilized as internal controls.

## Induction of osteogenic differentiation of bone marrow-derived MSCs and Alizarin Red staining

To detect the iron effect on osteogenic differentiation, Cyagen® osteogenesis differentiation medium (Cyagen Biosciences., Guangzhou, China) were used following the manufacturer's protocol (*Zhu, Mao & Gao, 2013*). Briefly, primary bone marrow-derived MSCs were cultured in osteogenesis differentiation media (2 mmol/L $\beta$-glycerol-phosphate, 50 μmol/L ascorbic acid, 0.1 μmol/L dexamethasone) alone or in the presence of FAC for 14 days. After induction, osteogenesis was eveluated by staining MSCs with Alizarin Red S reagent (Cyagen Biosciences., Guangzhou, China) as protocol described.

## Evaluation of the deposition of calcium

Primary bone marrow-derived MSCs were subcultured in 6-well plates in growth medium (Dulbecco's modified Eagle's medium, 10% fetal bovine serum (FBS) and 100 lg/mL streptomycin and penicillin). After primary bone marrow-derived MSCs had reached approximately 80% confluence, FAC was added to the osteogenesis differentiation medium. To quantify the iron effect on the matrix calcification, Alizarin red S staining were used following the manufacturer's protocol (*Zhang et al., 2010*). Positive red staining represents calcium deposits of on the differentiated MSCs. For quantification of staining, 100 mM cetypyridinium chloride was added to each plate and used to extract Alizarin red S, and

the sample was incubated at room temperature for 3 h (*Malladi et al., 2006*). Finally, the absorbance of the extracted Alizarin red S at 570 nm was analyzed in a spectrophotometer (Thermo, Waltam, MA, USA).

## Measurement of Alkaline phosphatase activity (ALP)

Primary bone marrow-derived MSCs treated on 6-well plates were rinsed with PBS thrice, lysed in RIPA solution (Beyotime Institute of Biotechnology, Jiangsu, China), and finally centrifuged to remove cellular debris at 4 °C. Next, ALP activity in the samples was measured by p-nitrophenyl phosphate method using a Alkaline Phosphatase Assay Kit (Beyotime Institute of Biotechnology, Jiangsu, China) as previous report (*Lyu et al., 2014*).

## Statistical analysis

All data were expressed as means $\pm$ standard deviation(SD).Statistical analyses were performed with SPSS software (version 18.0) for Windows software. In order to analyze statistical differences between samples, we used one-way analysis of variance (ANOVA) with least significant difference (LSD). $P < 0.05$ was considered statistically significant.

# RESULTS

## Influences of iron on osteoblastic cell viability

In our present study, the CCK-8 assay kit results indicated that iron had toxic effects. After a 120-h exposure to FAC, the viability of osteoblasts was found to be significantly inhibited by iron in a dose-dependent manner. However, after a 24-h exposure to FAC, no statistically significant difference was observed between the viability of osteoblasts at the various concentrations tested (Fig. 1). These results imply that iron may undergo accumulation in osteoblasts, which may elicit cytotoxic effects in these cells.

## Increase in intracellular labile iron levels due to iron overload in osteoblasts

Labile iron pool, known as free and chelatable iron, is the major potentially toxic form in iron-overload related diseases (*Brissot et al., 2012*). To evaluate changes of LIP in osteoblasts, a fluorescent iron-sensitive probe, calcein-AM was used. When calcein-AM permeates into the osteoblast and binds the intracellular labile iron, its fluorescence is quenched enabling evaluation of the intracellular labile iron levels by via measurement of the decrease in calcein-AM fluorescence (*Tenopoulou et al., 2007*; *Glickstein et al., 2005*; *Kaur et al., 2009*). After incubation with FAC (25–200 µM) for 120 h, the mean fluorescence intensity of calcein-AM decreased in a dose-dependent manner, which indicated a significant increase in the intracellular labile iron levels within the osteoblasts (Fig. 2).

## Impacts of iron on ROS level and expression of NADPH oxidase 4 (Nox4) in osteoblasts

To determine whether the over-production of ROS plays a pivotal role in iron-induced apoptosis, we used the specific fluorescence dye, H2DCF-DA, which detected intracellular ROS formation. According to the results of H2DCF-DA staining, a concentration-dependent increase in intracellular ROS production was observed in osteoblasts exposed

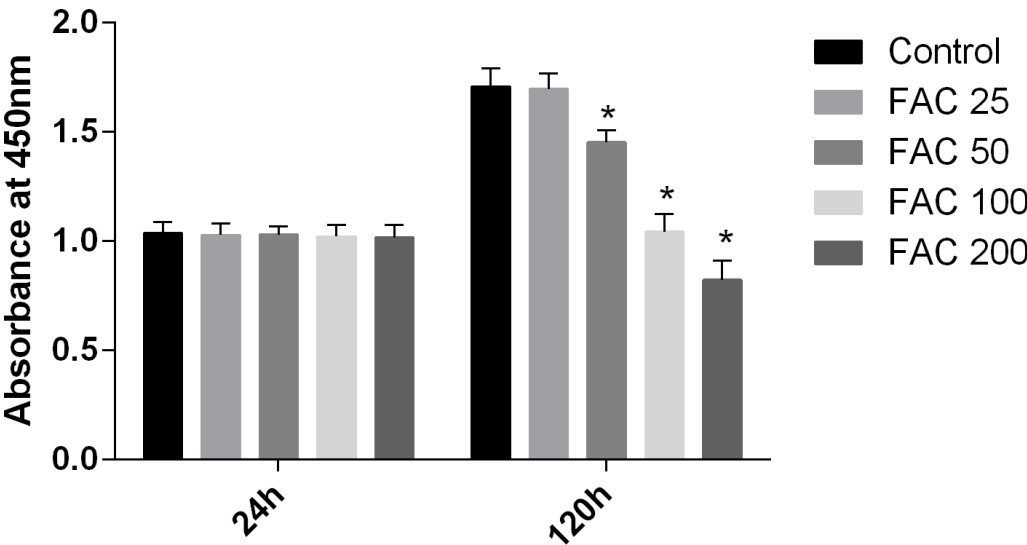

**Figure 1** **Cytotoxic effects of iron on the viability of osteoblasts.** Viability of osteoblasts was evaluated by CCK-8 assay after treatment with FAC (25–200 $\mu$M) for 24 h and 120 h. Compared to the control (FAC 0 $\mu$M), iron significantly reduced cell viability after 120-h FAC treatment. The values are presented as means $\pm$ SD, $n = 3$; *$P < 0.05$ vs. the control.

to various concentrations of FAC (25–200 $\mu$M) for 120 h (Fig. 3); the ROS levels were found to be 1.82- ,4.00- , 7.75-, and 10.55-fold higher after treatment of osteoblast with 25, 50, 100, and 200 $\mu$M FAC, respectively (Fig. 3B). NADPH oxidase is one of the most important ROS producers within the cell (*Sahoo, Meijles & Pagano, 2016*). In our study, we determined that FAC affects expression of Nox4, which mainly mediates the clinical phenotype in bone loss-related diseases (*Manolagas, 2010*; *Schröder, 2015*). Our results indicate that FAC (25–200 $\mu$M) upregulates Nox4 in osteoblasts.

## Effects of iron on apoptosis in osteoblasts

In order to assess whether iron mediated toxicity in osteoblasts is related to the activation of apoptosis, a following investigation was made. Annexin V/PI apoptosis assay kit were used to evlaute the apoptosis rate of osteoblasts. Annexin V+/PI- osteoblasts and Annexin+/PI+ osteoblasts were considered apoptotic cells. Following the application of FAC for 120 h at 0, 25, 50, 100 and 200 $\mu$M, apoptosis was seen to raise from 4.41% to 56.72%, as illustrated by (Fig. 4B). Furthermore, compared to control group (FAC 0 $\mu$M), there was a mainly raised amount of late apoptosis after FAC (200 $\mu$M) expouse in (Fig. 4A), with (Fig. 4C) also suggestive of a similar impact in osteoblasts.

Phase-contrast microscopy and fluorescence microscopy were also utilised to observe the morphology of osteoblasts and the degree of apoptosis resulting from iron exposure. The resluts of nuclear-staining with Hoechst indicated that, compared to 0 $\mu$M FAC (control), the proportion of osteoblasts with condensed or fragmented nuclei markedly increased in the FAC treatment group (Fig. 5A). Phase-contrast microscopy indicated that osteoblasts incubated with FAC (200 $\mu$M) for 120 h revealed apoptosis-related morphology, characterized by cell shrinkage, rounding and floating in 6-well plates (Fig. 5B).

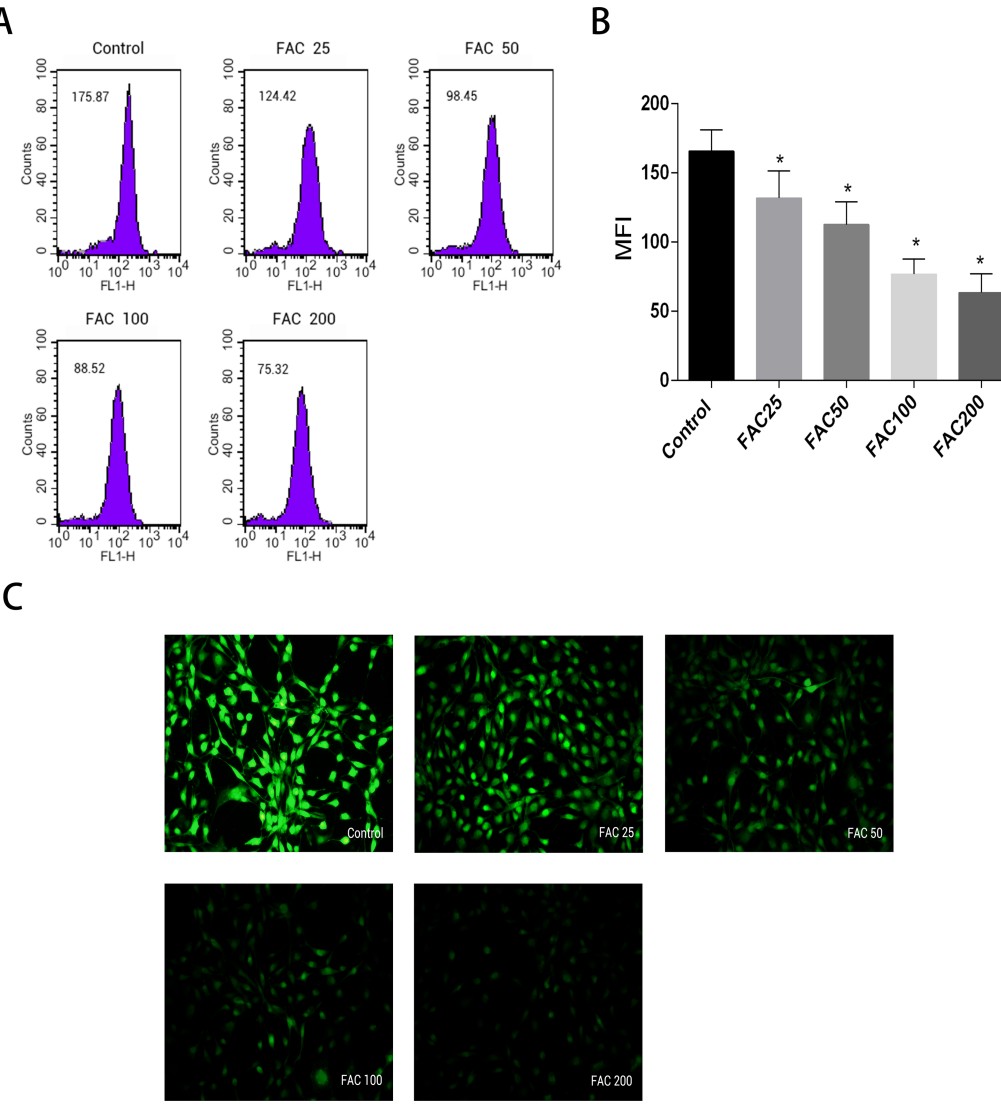

**Figure 2 Effect of iron-overload on the intracellular LIP in osteoblasts.** The intracellular LIP in osteoblasts markedly increased after treatment with 0–200 µM FAC for 120 h. (A) Representative flow cytometric results for intracellular LIP after FAC treatment. The intracellular LIP was estimated by calcein-AM, a fluorescent iron-sensitive probe. The probe fluorescence was quenched after chelating with labile iron; the mean fluorescence intensity (MFI) measured by flow cytometry was negatively correlated with intracellular LIP. (B) The reduction in MFI indicated an elevation in the intracellular LIP in the osteoblasts. Data are presented as the means $\pm$ SD, $n = 3$; *$P < 0.05$ vs. the control. (C) Representative fluorescence microscopy photomicrographs of intracellular LIP in osteoblasts. The quenching of green fluorescence indicates that the intracellular LIP was increased in cells.

## Involvement of cleaved Caspase-3, cytochrome c, Bax and Bcl-2 in iron-induced apoptosis

In order to confirm that the mitochondrial pathway associated with iron treatment, we detected the release of cytochrome c into cytoplasm and the generation of cleaved Caspase-3 by Western blot. As illustrated by Fig. 6, after FAC (25–200 µM) treatment for 120 h, a

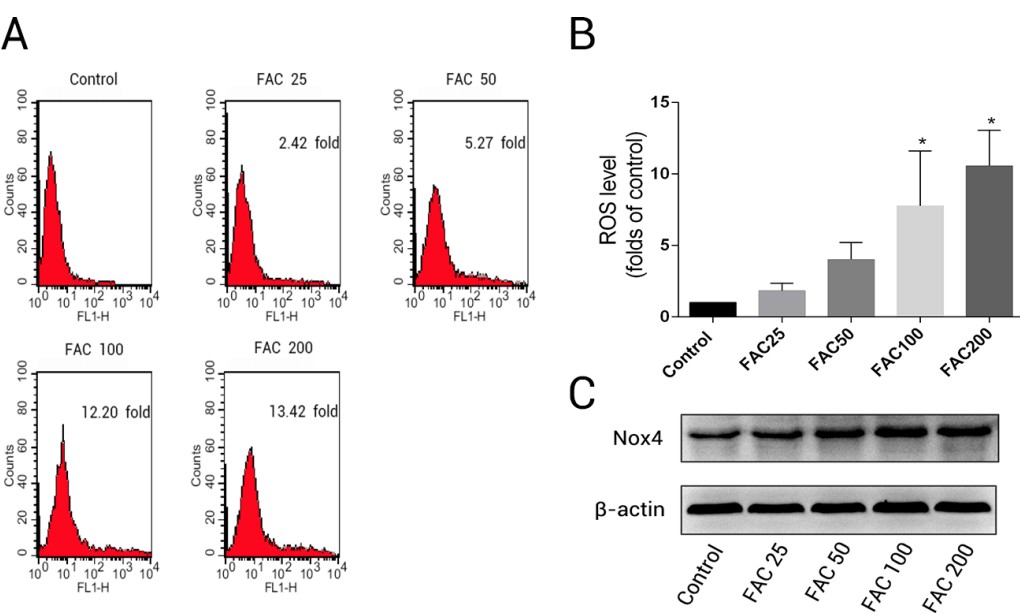

**Figure 3** **Iron induced ROS generation and upregulation of Nox4 in osteoblasts.** (A) Representative data of flow cytometric measurement of ROS production after labeling with H2DCF-DA. (B) Statistical bar graphs show the ROS levels in osteoblasts. The fluorescence intensity of osteoblasts was expressed relative to untreated osteoblasts (FAC 0 $\mu$M). Date are presented as means $\pm$ SD, $n = 3$; $^{*}P < 0.05$ vs. the control. (C) Representative western blot data for Nox4 in osteoblasts following exposure to FAC (0–200 $\mu$M) for 120 h. GAPDH was used as an internal control.

dose-dependent up-regulation of cytochrome c in this study was detected, which was accompanied by the generation of activated fragments of Caspase-3.

To investigate whether iron induces osteoblast apoptosis through alterations in Bcl-2 and Bax expression, Western blot were utilized to analyze Bax and Bcl-2 protein levels. In our present experiment, a dose-dependent down-regulation of Bcl-2, as well as an up-regulation of Bax was observed in osteoblasts exposed to FAC (25–200 $\mu$M) for 120 h (Fig. 6).

## Depolarisation of MMP in osteoblasts due to iron

Apoptosis could be initiated by the reduction of MMP, therefore it was decided to investigate whether such a reduction of MMP and consequent apoptosis is induced by iron. JC-1 dying of the osteoblasts and a flow cytometry were utilised to assess the amount of MMP. Figure 7 shows how MMP was diminished with the application of FAC in various quantities for 120 h. An increase in the strength of fluorescence of green JC-1 monomers compared to red JC-1 aggregates was considered to signify a significant reduction in MMP. Figure 7C illustrates how the control group's MMP had no reduction following JC-1 dying, due to apparent diminished strength of green fluorescence compared to red fluorescence. However, a breakdown of MMP was seen to be associated with apoptotic osteoblasts caused by increased iron levels. An increased number of mitochondria were observed with a greater amount of green fluorescence, following the addition of FAC at varying concentrations of 25–200 $\mu$M, for a duration of 120 h.

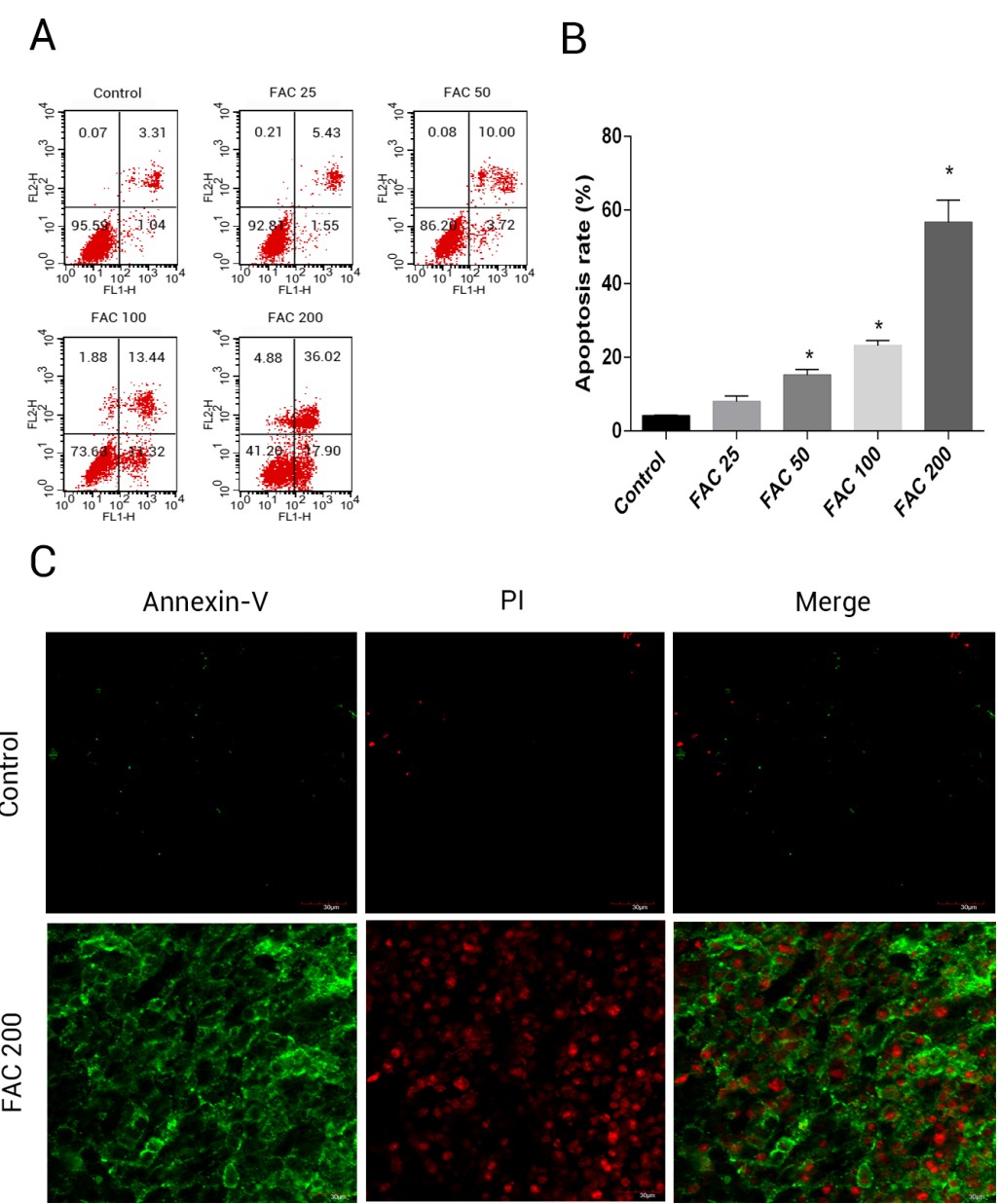

**Figure 4   Iron induced apoptosis in osteoblasts.** (A) Representative flow cytometric analysis of apoptotic osteoblasts stained for Annexin V/PI after exposure to 0–200 µM FAC for 120 h. In each plot, the lower left quadrant represents live osteoblasts, the lower right and upper right quadrants represent apoptotic osteoblasts, and the upper left quadrant represents necrotic osteoblasts. (B) Statistical bar graphs show the mean values of flow cytometry data. Data are presented as the means ± SD, $n = 3$. *$P < 0.05$ vs. the control. (C) Representative photomicrograph (OLYMPUS FV1000; Olympus, Tokyo, Japan) of osteoblasts stained with AnnexinV/PI dye after treatment with 0 µM and 200 µM FAC for 120 h. Apoptotic osteoblasts were defined as annexin-V+/ PI- cells and annexin-V+/PI- cells.

## N-acetyl-L-cysteine (NAC)'s protection impact on iron-related apoptosis in osteoblasts

This was to ascertain the extent to which NAC can prevent apoptotic occurrences related to iron in osteoblasts, by reducing the creation of ROS. The results suggested that the

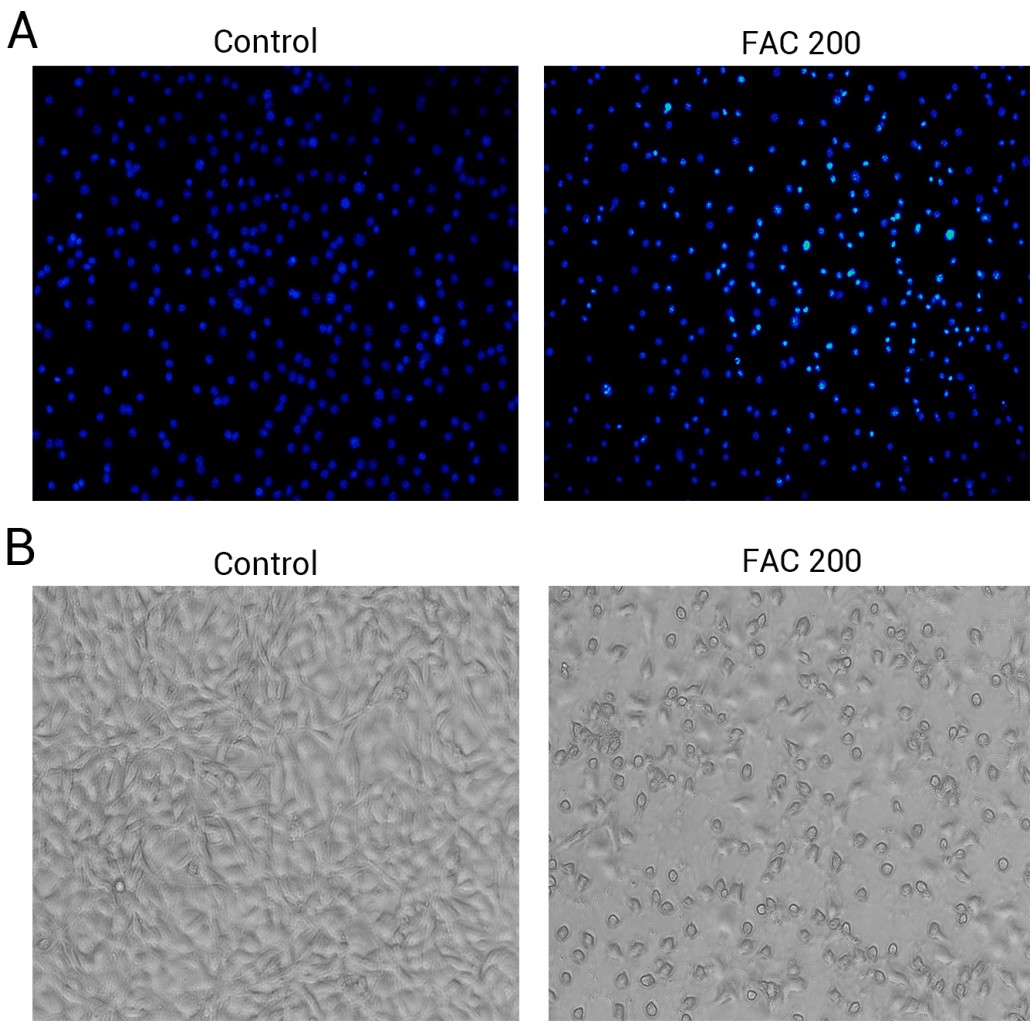

**Figure 5  The morphological changes of apoptosis in osteoblasts FAC.** (A) Hoechst 33258 staining of osteoblasts after treatment with PBS (Control) and 200 $\mu$M FAC (FAC 200) for 120 h. Apoptotic osteoblasts showed condensed and bright nuclei stained by Hoechst 33258. (B) Phase-contrast photomicrograph of osteoblasts after treatment with PBS (Control) and 200 $\mu$M FAC (FAC 200) for 120 h. Apoptotic osteoblasts presented shrinkage and swelling and detached from the plates.

application of NAC was able to reduce the mitochondria's loss of cytochrome c, reduce the creation of ROS as an effect of iron to a large degree and consequently diminish the breakdown of MMP related to iron (Fig. 8). As illustrated by Fig. 8E, Caspase-3's stimulation was thus inhibited by NAC, Bcl-2/Bax modulation was inverted, while apoptotic osteoblasts resulting from the effects of iron were effectively reduced by NAC's application to osteoblasts. Therefore, apoptosis as a result of the effects of iron can also be seen to be significantly exacerbated by ROS production.

## Effects of iron on bone marrow-derived MSCs viability and apoptosis

MSCs, characterised by their multipotent differentiation capacity, are recruited to the bone remodeling surface and differentiated into osteoblasts, which play an essential role to

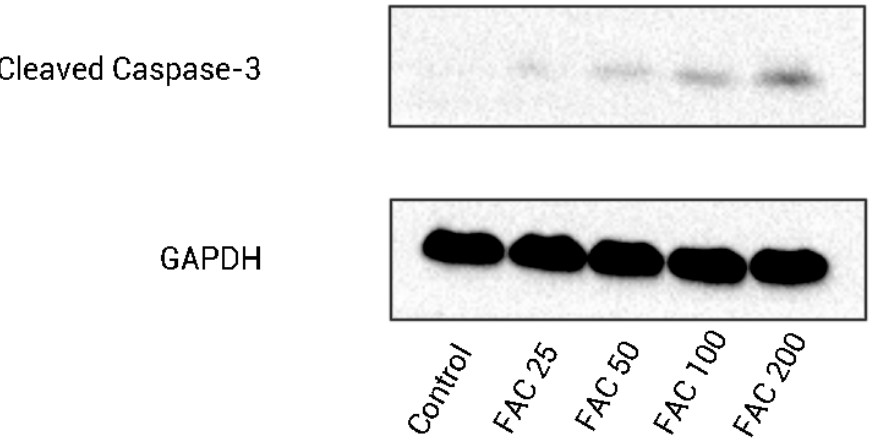

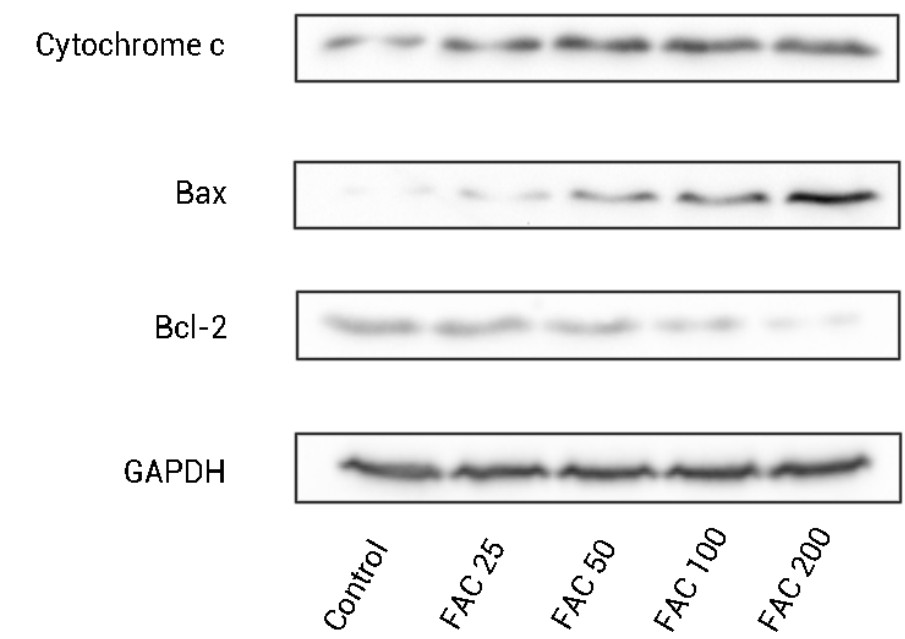

**Figure 6** **The expression of apoptosis-related proteins in osteoblasts.** Representative western blot data for cleaved Caspase-3, Bax, Bcl-2, and cytosolic cytochrome c in osteoblasts following exposure to FAC (0–200 $\mu$M) for 120 h. GAPDH was used as an internal control.

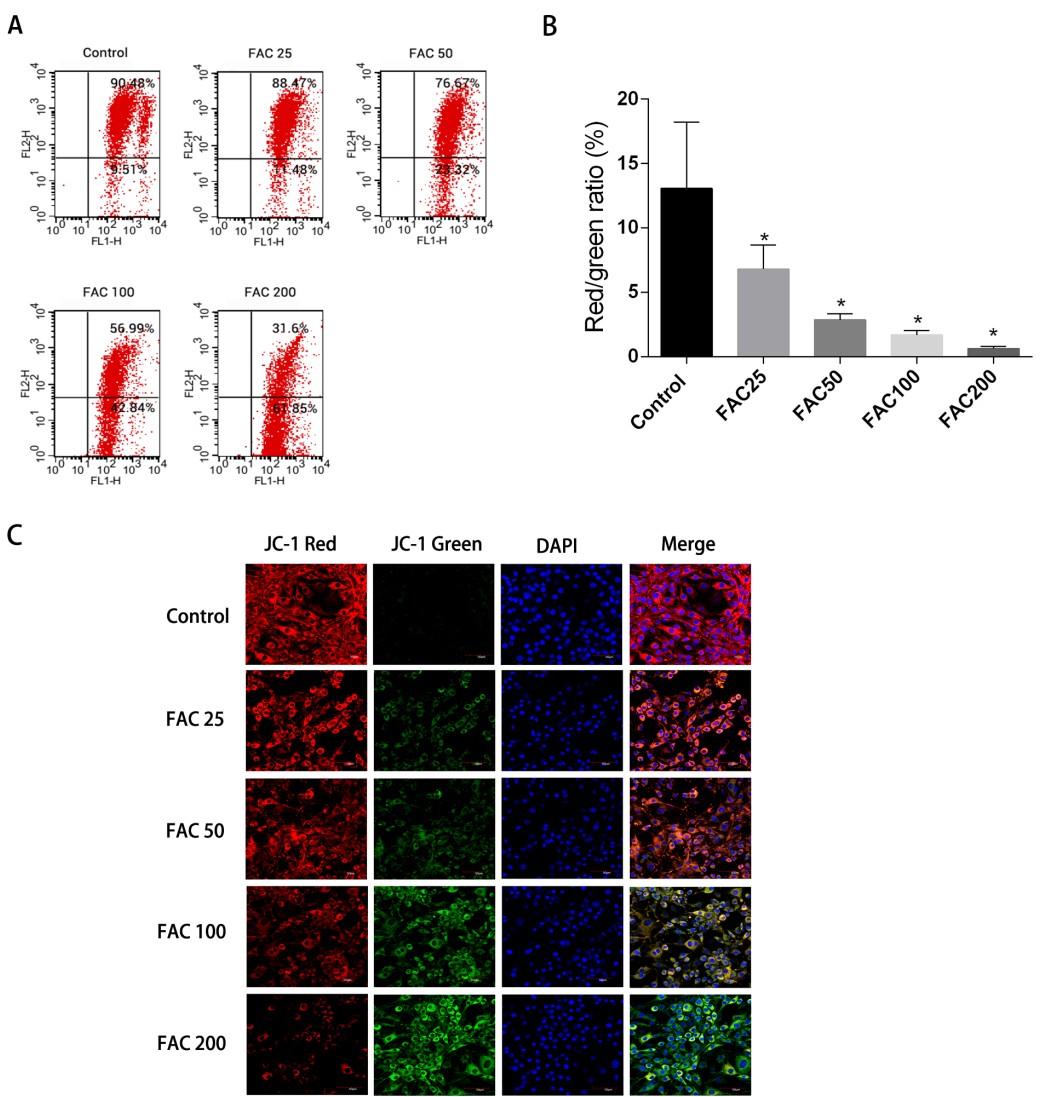

**Figure 7** **Iron-induced decrease in MMP of osteoblasts.** The MMP in osteoblasts treated with FAC (0–200 µM) for 120 h as measured by JC-1 staining. (A) Representative graphs of flow cytometric analysis of the altered MMP after incubating with JC-1 dye. (B) Statistical bar graphs show the changes of MMP detected by flow cytometry. The changes of MMP in osteoblasts were defined as the ratio of red/green fluorescence intensity. Data are presented as the means ± SD, $n = 3$. *$P < 0.05$ vs. the control. (C) Representative laser scan confocal microscopy photomicrographs of osteoblasts stained with JC-1 dye.

maintain bone mass. To test the effects of iron on the viability of MSCs *in vitro*, the CCK-8 assay kit was used in our study. As shown in Fig. 9, iron inhibited the viability of MSCs at 72 h and 144 h in a dose-dependent manner. Next, we detected whether apoptosis was involved in iron-induced cellular toxicity. As reported in Fig. 10, a significant increase of the apoptosis rates was observed in bone marrow-derived MSCs exposed with FAC (200 µM) for 144 h. Taken together, iron overload-induced toxicity of bone marrow-derived MSCs appears to be partially mediated by apoptosis.

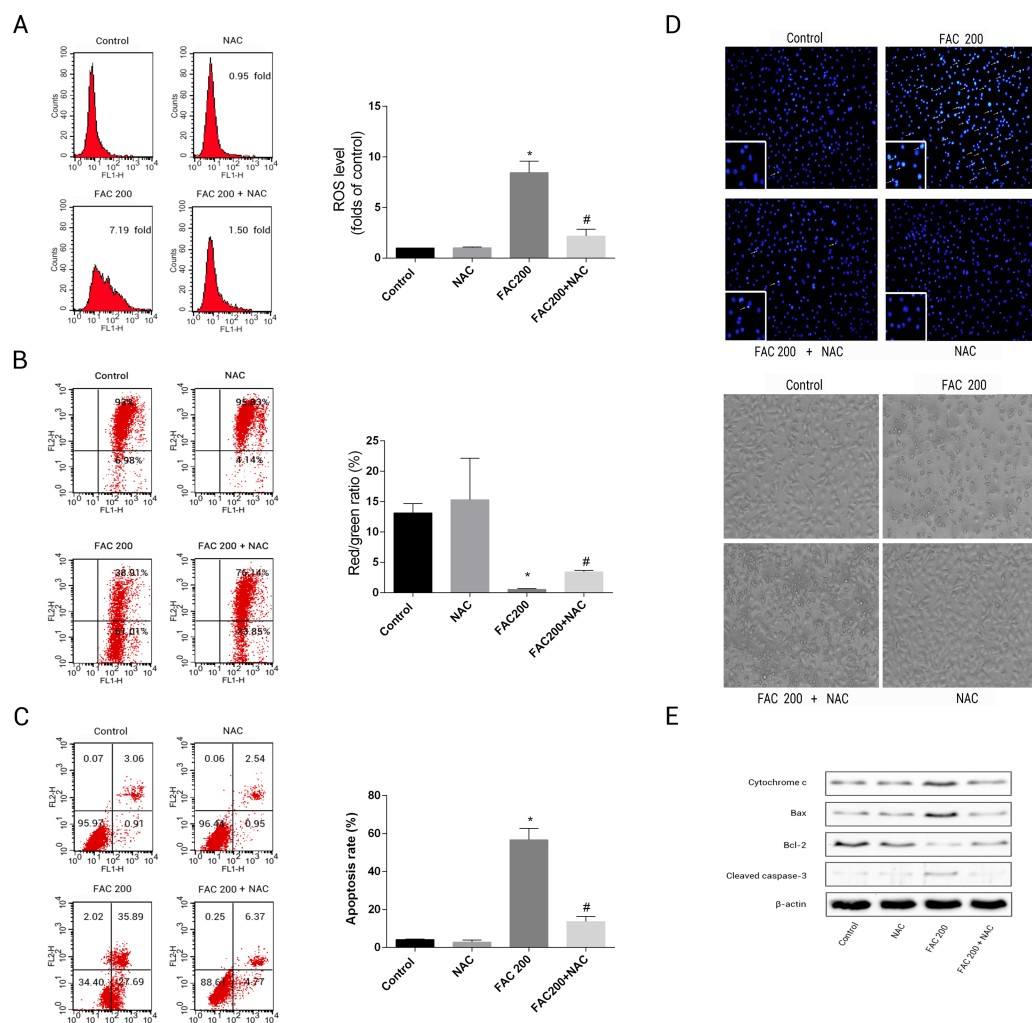

**Figure 8** **Protective effects of NAC against iron-induced apoptosis.** Osteoblasts were exposed to FAC (200 μM) with or or without NAC (1 mM) for 120 h. (A) The change of intracellular ROS levels in osteoblasts treated with FAC (200 μM) with or without NAC (1 mM) for 120 h. *$P < 0.05$ vs .the control; #$P < 0.05$ vs. FAC 200. (B) The change of MMP in osteoblasts incubated wtih FAC (200 μM), in the absence or presence of 1 mM NAC for 120 h, as assayed by flow cytometry. *$P < 0.05$ vs. the control; #$P < 0.05$ vs. FAC 200. (C) The effect of NAC on iron-induced cell apoptosis as assayed by flow cytometry analysis. *$P < 0.05$ vs. the control; #$P < 0.05$ vs. FAC 200. (D) The effect of NAC on iron-induced morphological changes in cells as visualized by phase-contrast micrograph and Hoechst 33342 staining. (E) The effect of NAC on the expression of apoptosis-related proteins in iron-treated osteoblastic cells.

## Inhibitory effects of iron on osteogenic differentiation and mineralization

To investigate whether excess iron impaire osteogenesis of bone marrow-derived MSCs and mineralization, cells were cultured in osteogenesis differentiation media alone or in the presence of FAC (25–200 μM) for 14 days. Next, the activity of ALP, a specific marker of osteogenic differentiation, was detected in bone marrow-derived MSCs. Figure. 11A shows that iron caused a concentration-dependent inhibitory effect of the activity of ALP in osteogenesis of MSCs. In addtion, Alizarin Red staining showed that iron leaded to a

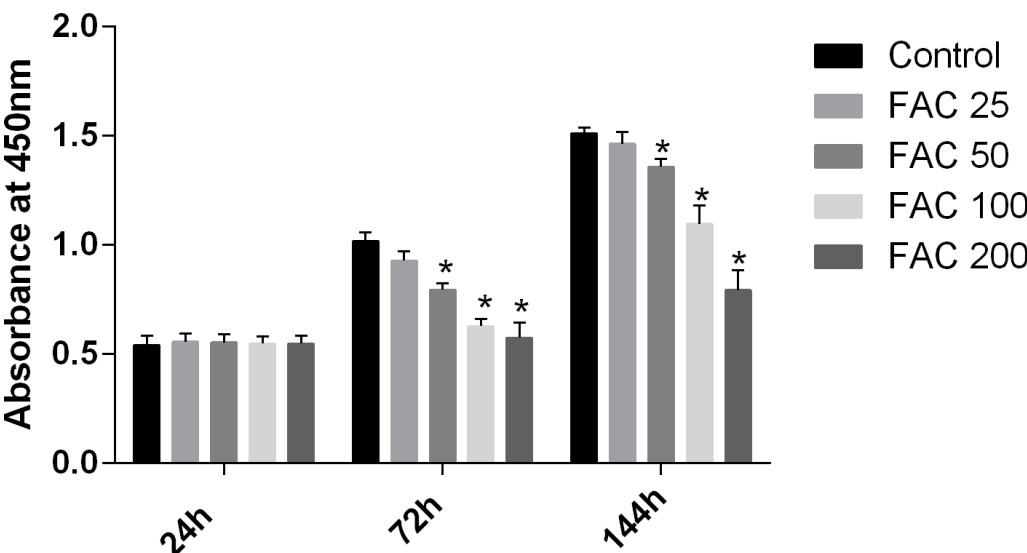

**Figure 9  Cytotoxic effects of iron on the viability of bone marrow-derived MSCs.** Viability of bone marrow-derived MSCs was evaluated by CCK-8 assay after treatment with FAC (25–200 μM) for 24 h, 72 h and 144 h. Compared to the control (FAC 0 μM), iron significantly reduced cell viability after 72 h and 120 h FAC treatment. The values are presented as means ± SD, $n = 3$; *$P < 0.05$ vs. the control.

concentration-dependent impairment of mineralization. At the concentration of 200 μM, iron almost completely inhibited the mineralization process of MSCs *in vitro*. Then, we estimated the effect of iron-overload on calcium deposotion of the extracellular matrix. As demonstrated in Figs. 11B–11D, FAC caused a decrease in the calcium content of the extracellular matrix in a dose-dependent manner, which is in a accordance with the Alizarin Red staining results.

## DISCUSSION

Osteoporosis has been reported to be closely related with iron overload, which may arise due to conditions such as hemochromatosis and the thalassemia (*Wong et al., 2013*; *Haidar, Musallam & Taher, 2011*; *Valenti et al., 2009*). Meanwhile, various studies confirmed that age-associated iron accumulation is a contributing factor in the pathogenesis of postmenopausal osteoporosis (*Li et al., 2012*; *Fiona , 2012*). It has also been demonstrated that increased physiological iron stores could accelerate bone loss, even in healthy adults (*Kim et al., 2012*; *Fiona , 2012*). The maintenance of normal skeletal homeostasis relies on osteoblasts, which is responsible for bone matrix synthesis, secretion and mineralization. The impairment of osteoblasts is considered to be a major factor contributing to osteoporosis in iron overload related diseases (*Domrongkitchaiporn et al., 2003*; *Terpos & Voskaridou, 2010*; *Mahachoklertwattana et al., 2003*; *Doyard et al., 2016*). Therefore, it is essential to investigate the toxic effects of iron on osteoblasts and elucidate the molecular mechanisms of iron toxicity in these cells. Our present experiment reveals that iron-overload causes deterimental effects in the osteoblastic cellular proliferation. We have also found that iron-overload effectively induces apoptosis in osteoblasts *in vitro*, which is in

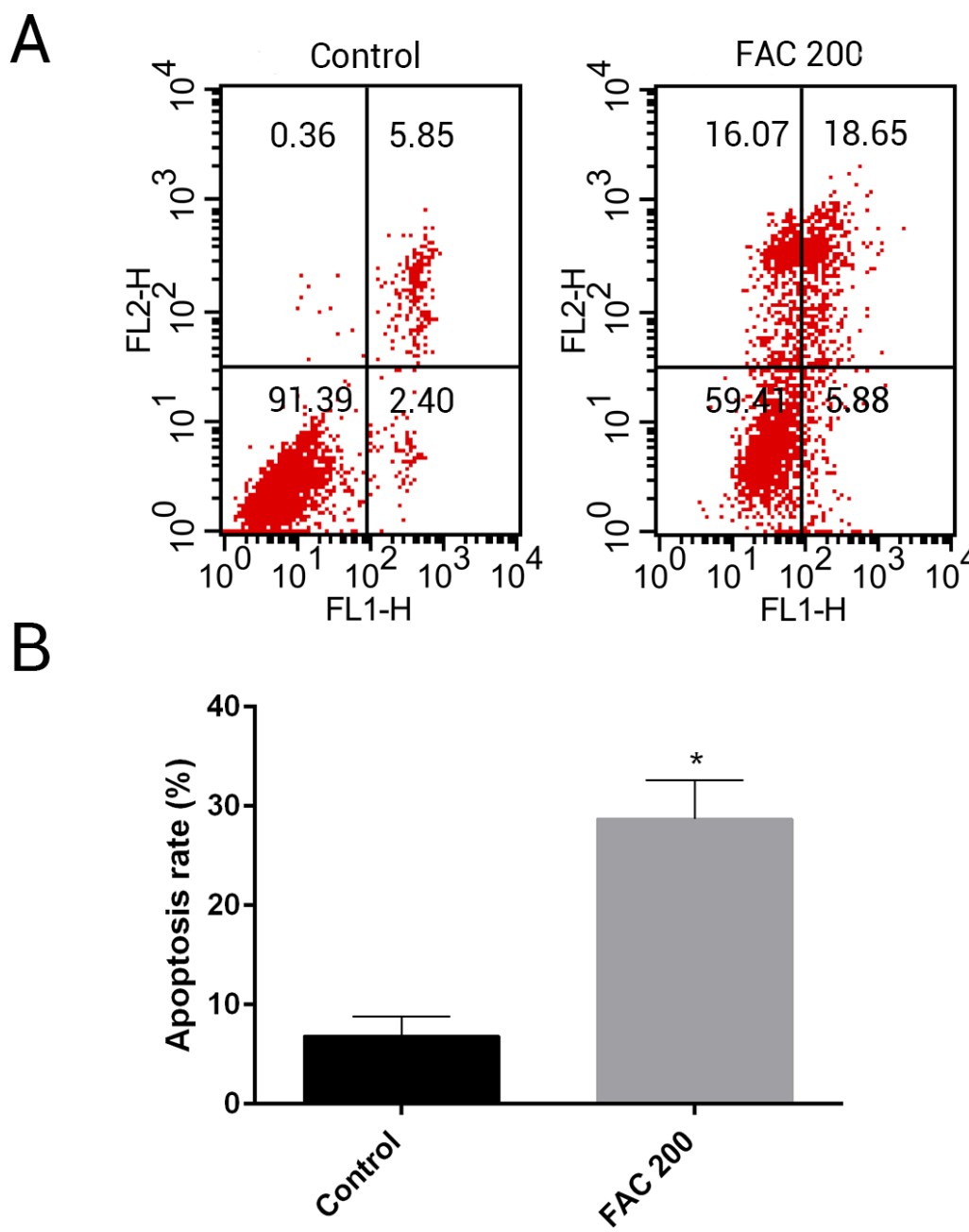

**Figure 10  Iron induced apoptosis in bone marrow-derived MSCs.** (A) Representative flow cytometric analysis of apoptotic MSCs stained for Annexin V/PI after exposure to 200 µM FAC for 144 h. In each plot, the lower left quadrant represents live MSCs, the lower right and upper right quadrants represent apoptotic MSCs, and the upper left quadrant represents necrotic MSCs. (B) Statistical bar graphs show the mean values of flow cytometry data. Data are presented as the means $\pm$ SD, $n = 3$. *$P < 0.05$ vs. the control.

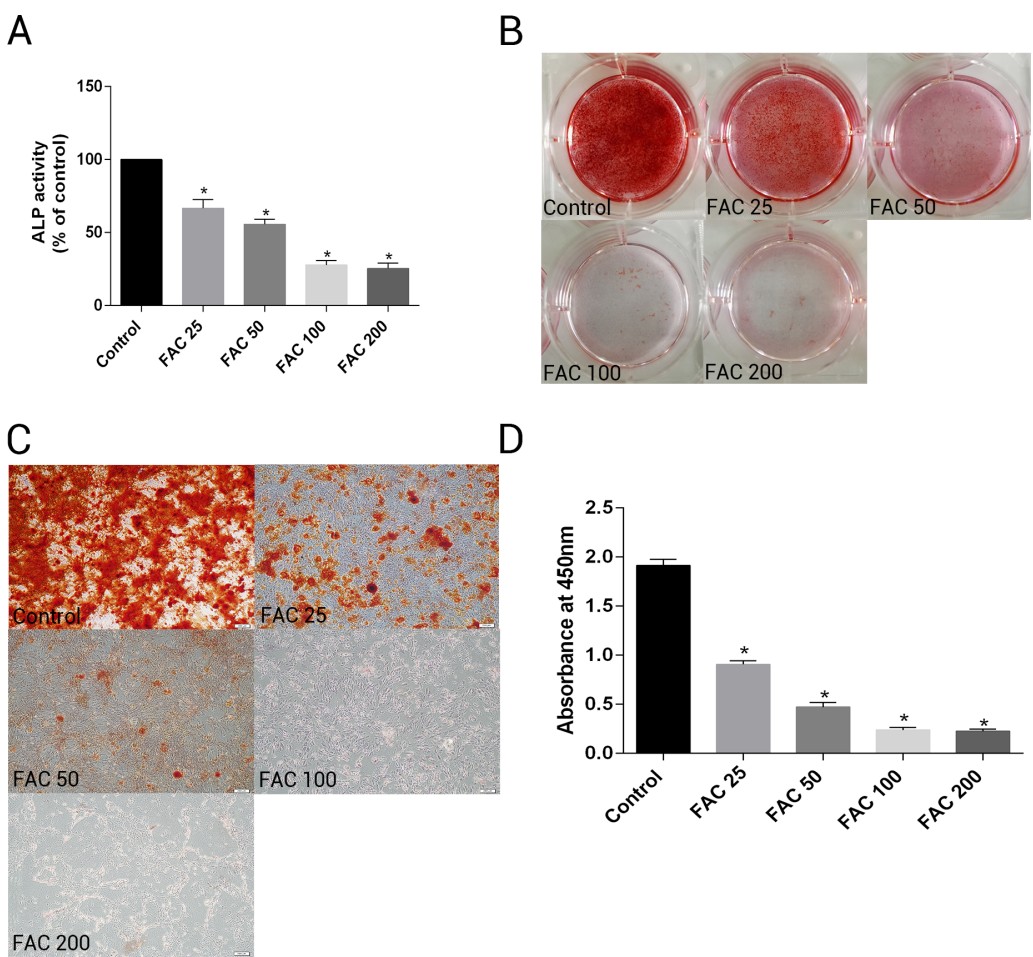

**Figure 11  Effect of iron on ALP activity and matrix calcification.** (A–D) Primary bone marrow-derived MSCs cultured in 6-well plates were induced with osteogenic medium supplemented with FAC (25–200 μM) or alone for 14 d. (A) ALP activity of MSCs was detected as described in Materials and Methods. Data are presented as the means ± SD, $n = 3$. $*P < 0.05$ vs. the control. (B–C) Mineralization in bone marrow-derived MSCs was assayed by Alizarin Red staining. Representative photographic images of stained wells (B) and microscopic views (C) are shown. (D) Statistical bar graphs show Ca content from different groups. Data are presented as the means ± SD, $n = 3$. $*P < 0.05$ vs. the control.

accordance with the findings of previously published research (*Messer et al., 2009*; *Doyard et al., 2012*). More interestingly, our study, for the first time, suggests that the apoptosis caused by iron overload is correlated with activation of the mitochondrial pathway.

While iron is a fundamental element for various crucial biological processes, such as enzymatic reactions and oxygen transport, excess iron accumulation could damge cells by catalyzing the over-production of damaging hydroxy radical through Haber–Weiss reactions (*Dixon & Stockwell, 2014*). Labile iron pool, known as free and chelatable iron, is the major potentially toxic form in iron-overload related diseases (*Esposito et al., 2003*; *Devos et al., 2014*; *Berdoukas, Coates & Cabantchik, 2015*; *Chai et al., 2015*). In this study, we observed that the intracellular LIP level in osteoblasts was significantly increased after exposure to FAC for 120 h. ROS, which induced by labile iron via iron-dependent oxidase

(for example, xanthine oxidase, nicotinamide adenine dinucleotide phosphate hydride oxidases, lipoxygenases) or the Haber–Weiss reaction, is primary responsible for cellular damage caused by iron overload (*Dixon & Stockwell, 2014*). Numerous studies reported that celluar and genomic oxidative damage were highly correlated with elevated levels of labile iron in thalassemia patients (*Brissot et al., 2012*; *Berdoukas, Coates & Cabantchik, 2015*). Next, we evaluated the generation of ROS in osteoblasts after exposure to FAC and found that iron-overload induced ROS production drastically increased. Moreover, the over-production ROS was paralleled with the increase of intracellular labile iron and the cytotoxicity of osteoblasts. The generation of ROS is tightly regulated through various ways, including NADPH oxidases, phagocyte oxidase and mitochondrial electron transport chain (*Sahoo, Meijles & Pagano, 2016*; *Dikalov, 2011*; *Zhou et al., 2013*). Emerging evidence indicates that NADPH oxidases are the primary generator of ROS in the skeletal system and Nox-derived ROS are key players which mediates osteoblasts dysfunction with osteoporosis (*Manolagas, 2010*; *Schröder, 2015*). In our study, we found that iron overload could upregulate the expression of NADPH oxidase 4 (Nox4) in osteoblasts after treatment with FAC. In addition, heme (an iron derivative), as electron transporter, plays an important role for superoxide generation of NOX family NADPH oxidases (*Bedard & Krause, 2007*). Therefore, Nox4 may be an essential part in iron-overload induced generation of ROS in osteoblasts.

Previous studies have also shown that iron overload exerts deterimental effects on various cell types and its mechanisms by which iron toxicity occurs are closely associated with apoptosis (*Chan, Yu Ye & Chan, 2013*; *Park et al., 2015*; *Dussiot et al., 2014*). In this experiment, we demonstrated that iron-overload effectively induced apoptosis in osteoblasts. Furthermore, the activation of Caspase-3 was also observed after treatment with FAC. Although it is well established that iron-overload could induce apoptosis, its exact pathway in osteoblasts is still largely unknown. In iron-overload conditions, excess labile iron enters the mitochondria via the calcium uniporter, and then interacts with reactive oxygen intermediates leaked from mitochondrial respiratory chain through Fenton reactions, catalyzing powerful ROS to damage mitochondria (*Chen et al., 2014*; *Sripetchwandee et al., 2014*; *Pelizzoni et al., 2011*; *Uchiyama et al., 2008*). ROS catalyzed by labile iron elicits a range of detrimental effects in mitochondria, as following (1) impairment of mitochondrial respiratory enzyme activity; (2) decrease in ATP production; (3) loss of MMP; and (4) damage to mitochondrial DNA (mtDNA) (*Ward et al., 2014*; *Mallikarjun et al., 2014*; *Al-Qenaei et al., 2014*; *Santambrogio et al., 2015*; *Rouault, 2016*; *Rines & Ardehali, 2013*). In this experiment, the inhibition of mitochondrial dehydrogenase activity was observed in osteoblasts exposed to FAC. Considering that ROS enhanced by iron overload could impair mitochondrial ultrastructure and disrupt its function, which might subsequently activate Caspase-3 through various molecular cascade reactions, we hypothesized that iron-overload induced apoptosis of osteoblasts might occur through the mitochondrial pathway.

The mitochondrial apoptosis pathway has been well documented by many previous studies (*Czabotar et al., 2014*). In the mitochondrial pathway, mitochondrial membrane permeabilization, characterized by the loss of MMP, is considered as the most critical

event activating caspase and causes apoptosis (*Fuchs & Steller, 2011*; *Kroemer, Galluzzi & Brenner, 2007*). After mitochondrial membrane permeabilization induced by various apoptotic stimuli, Cyto c releases from mitochondrial intermembrane space, subsequently binds APDF1 in cytosol, and then forms apoptosome through recruiting and activating caspase 9. Eventually, caspase-9 cleaves and activates caspase-3, resulting in the activation of apoptotic cell death (*Fuchs & Steller, 2011*; *Kroemer, Galluzzi & Brenner, 2007*). Therefore, the changes of MMP in osteoblasts after treatment with FAC were studied in details. Confocal microscopy observation indicated that iron overload led to a dose-dependent decrease of MMP in osteoblasts. Furthermore, the depolarization of MMP was subsequent accompanied with Cyto C release from mitochondria into the cytoplasm. Based on above data, we then further to study the expression changes of other essential molecules regulating the mitochondrial apoptosis pathway to prove our hypothesis. Bcl-2 family proteins have been confirmed to control cellular apoptosis by directly or indirectly regulating mitochondrial membrane permeabilization (*Czabotar et al., 2014*). Bcl-2, an anti-apoptotic molecule, and Bax, an pro-apoptotic molecule, are key members among the Bcl-2 family (*Hardwick & Soane, 2013*). The decrease in Bcl-2 or increase in Bax, could promote permeabilization of the mitochondrial membrane, leading to the release of Cyto c and eventually triggering apoptosis (*Kroemer, Galluzzi & Brenner, 2007*). In our study, we observed that iron overload caused the upregulation of Bax and cleaved caspase-3, as well as the downregulation of Bcl-2. The changes of Bcl-2 and Bax expression in osteoblasts may have been sufficent to facilitate mitchondrial membrane permeability. Taken together, our findings indicate that the mitochondrial apoptosis pathway might be invovled, at least in part, in iron overload-releated osteoblast injury.

NAC is a well-known antioxidant, which elevates the intracellular glutathione levels, an important module in the cellular antioxidative system (*Tsay et al., 2010*). In our study, we found that apoptosis induced by iron overload in osteoblasts was associated with increased harmful free radicals and was also largely prevented by NAC. Meanwhile, previous studies also found that NAC could enhance osteogenesis and inhibit osteoclast differentiation (*Yamada et al., 2013*; *Jun et al., 2008*; *Hyeon et al., 2013*; *Lee et al., 2005*). This suggests that NAC could be an adjunctive therapy in iron-overload related bone loss. The concept that increased harmful free radicals induced by iron overload is the main contributer of iron toxicity is not new. But, to our knowledge, detailed mechanism for NAC protection against iron overload-induced osteoblasts apoptosis has not been reported. Our findings revealed that NAC could prevent the mitochondria damage casused by iron overload through directly scavenge the over-generated ROS. Thus, cytochrome c released from mitochondria was decrease and the the activation of caspase-3 was inhibited. In addition, after NAC treatment, the expression of Bcl-2 was markedly increase, while the expression of Bax was decrease. This result might imply that Bcl-2 family proteins also involved in the NAC protection effects.

Our study was highly reproducible; however, some limitations should be noted. Firstly, our studies were conducted *in vitro*; as such our results may not be representative of biological process *in vivo*. Furthermore, as human osteoblastic cells are difficult to obtain, we utilized the MC3T3-E1 osteoblastic cell line to examine the toxicity of iron

in this study. Although numerous studies have reported that the MC3T3-E1 cell line is similar to human osteoblasts in function (*Czekanska et al., 2012*), further studies using human osteoblasts are warranted. Finally, our results indicate that the mitochondrial apoptotic pathway is involved in mediating iron toxicity in osteoblasts. However, the iron-mediated destabilization of lysosomal membranes represents an alternative mechanism of iron toxicity. In future experiments, we aim to explore the potential effects of iron on lysosomes, which may include, lysosomal membrane permeabilization and cross-talk between lysosomes and mitochondria.

In the maintenance of skeletal homeostasis, besides osteoblast, mesenchymal stem cell also also plays an essential role in osteogensis. It has been reported that both the number and osteogenic differentiation potential of bone marrow-derived MSCs decrease in osteoporotic patients (*Xian et al., 2012*; *Guan et al., 2012*). In our experiments, we found that iron caused a concentration-dependent inhibitory effect of the viability of bone marrow-derived MSCs. Furthermore, iron overload in bone marrow-derived MSCs result in increased apoptosis. This is similar to our results in osteoblasts and also consistent with previous reports (*Chai et al., 2015*; *Zhang et al., 2015*; *Lu et al., 2013*). To explore the effect of iron-overload on the osteogenic differentiation of bone marrow-derived MSCs, we estimated the change of ALP activity. In response to osteogenic induction, bone marrow-derived MSCs could increase the activity of ALP, a specific marker of osteogenic differentiation. In iron-overload condition, this response was significantly attenuated. In addition, iron could directly inhibit matrix mineralization of bone marrow-derived MSCs. Numerous clinical and *in vivo* studies have also indicated that defective mineralization of bone was one of the pathological changes in iron overload-related osteoporosis (*Doyard et al., 2016*; *Mahachoklertwattana et al., 2003*; *Matsushima et al., 2003*). Our *in vitro* findings that excess iron caused MSCs apoptosis and impaired osteogenic differentiation and mineralization might, at least in part, offer understanding of low bone density in iron overload diseases.

Overall, our data indicate that iron significantly induces apoptosis in osteoblasts *in vitro*. NAC could remarkably relieve iron overload-induced osteoblasts apoptosis. In addition, we demonstrate that iron induces apoptosis via the enhanced production of ROS, which impairs mitochondrial function and leads to MMP collapse, cytochrome c release, and caspase activation. This provides valuable insights into the molecular mechanisms underlying osteoblastic cell death in the iron-overload condition. Meanwhile, we also revealed that iron overload could promote apoptosis and impair osteogenic differentiation and mineralization in bone marrow-derived MSCs.

### Funding
The authors received no funding for this work.

### Competing Interests
The authors declare there are no competing interests.

## Author Contributions

- Qing Tian conceived and designed the experiments, performed the experiments, analyzed the data, contributed reagents/materials/analysis tools, wrote the paper, prepared figures and/or tables.
- Shilei Wu performed the experiments.
- Zhipeng Dai analyzed the data, reviewed drafts of the paper.
- Jingjing Yang analyzed the data, contributed reagents/materials/analysis tools, wrote the paper.
- Jin Zheng analyzed the data, reviewed drafts of the paper.
- Qixin Zheng and Yong Liu conceived and designed the experiments.

## Data Availability

The raw data has been supplied as Data S1.

## Supplemental Information

Supplemental information for this article can be found online at http://dx.doi.org/10.7717/peerj.2611#supplemental-information.

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
