# Peer review of "Iron overload induced death of osteoblasts in vitro: involvement of the mitochondrial apoptotic pathway"

_PeerJ, doi:10.7717/peerj.2611_

## Round 0.1 · original submission · Major Revisions

Both reviewers find your work interesting and have several suggestions. Please, read through them carefully and although in vivo experiments (Reviewer 1) would be very time consuming, I feel that the suggested in vitro experiments (Reviewer 2) should be done. These additional data will elevate the value of the work significantly.

Reviewer 1 ·

Basic reporting

There is sufficient background with Figures that are easy to follow and interpret.

Experimental design

The design used classic dose response with a well characterized osteoblast cell line. The rationale for the study was clearly outlined and conducted rigorously.

Validity of the findings

The data appear controlled, statistically sound with the correct conclusions.

Additional comments

The paper examines the effects of iron toxicity on osteoblast functions. The scientific basis for this work comes from previous work showing patients with conditions of iron overload (as in thalassemia or hemochromatosis) have a higher incidence of osteoporosis (low bone mass) and fractures. The authors use a well characterized mouse osteoblastic cell line known as MC3T3-E1 to first carry out dose response to cytotoxicity and establish a dose response curve that is used throughout the paper (from 25 umolar to 200 umolar FAC). The MC3T3 were then exposed to different doses of FAC used to test the effects of iron (in the form of FAC) on intracellular LIP, ROS and Nox4 generation, Annexin V/PI, nuclear changes with Hoechst 33258, cell shrinkage and cell detachment. The expression of apopotisis related genes including BAX, Bcl-2 and caspace 3 all showed a dose dependent increase from iron exposures. The FAC also caused a decrease in MMP. Many of the observed changes in MC3T3-E1 outlined above could be reversed by application of NAC. While it would have been nice if an in vivo approach could be used to compliment the in vitro work described in the paper, the experiments that are shown are convincing and worth documenting. There were a number grammatical issues that are outlined as follows that the authors should review and revise in order to strengthen their presentation:

Line 137 change “certificate” to “verify”
Line 141, change “two cleans” to “two rinses/washes”
Line 142, its not needed to explain that Cal Zeiss is a German company
Line 227 change “instigated” to “induced”
Line 232 remove “suffered”
Line 237, rephrase “constraining”
Line 240 and 244 change “constrain/contrained” to “reduce/reduced”
Line 262 change “researches” to “research”
Line 284 change “osteoblast dysfunction under” to “osteoblast dysfunction with”
line 304 rephrase “detrimental”
Line 317 change “leaded” to “led”
Line 318 change “were” to “was”

Reviewer 2 ·

Basic reporting

Overall the paper is well-written, with sufficient introduction and background information.

Experimental design

Tian at al sets out to investigate the effect of iron-overload on osteoblast apoptosis in a cell line in vitro. Although the study is well-designed and excecuted, there are a few additional experiments that would should be added before acceptance for publication. See below.

Validity of the findings

The data presented here is statistically sound and controlled.

Additional comments

1. Please provide a time course between 24 and 120 hours (48, 72, 96).
2. The authors speculate that apoptosis induced by iron in this in vitro assay may serve as an explanation for osteoporosis in chronic iron overload states. I find this a bit of a stretch. The changes seen in vitro are rather acute compared to chronic exposure to soluble iron in an in vivo setting over the course of several years. Although not perfect, a better model may be to see how chronic (4-8 weeks) addition of lower level of iron to the cell culture may effect cell survival, CFU efficiency etc.
3. It would be interesting to see how iron effects primary mesenchymal stem cell derived osteoblast survival.
4. An even more relevant experiment would be to explore the effect of iron on bone formation in vitro. Alizarin-red staining may be quantified.
4. The authors explore the role of NOX4 in iron mediated oxidative stress. Any role for dual oxidases (DUOX)?

---

## Round 0.2 · accepted · Accept

Thank you for responding to the reviewers' suggestions and resubmitting your work. The changes and responses significantly improved the submission.

Reviewer 1 ·

Basic reporting

Adheres to guidelines

Experimental design

Design clearly stated

Validity of the findings

Findings are sound and controlled

Additional comments

The authors have adequately addressed this reviewers comments.

Reviewer 2 ·

Basic reporting

The article has improved substantially with the added new data.

Experimental design

No comments.

Validity of the findings

No comments.